# The Hyperloop System and Stakeholders: A Review and Future Directions

**Lambros Mitropoulos \*, Annie Kortsari, Alexandros Koliatos and Georgia Ayfantopoulou**

Centre for Research and Technology Hellas, Hellenic Institute of Transport, 57001 Thessaloniki, Greece;
akorts@certh.gr (A.K.); alexkoliatos@certh.gr (A.K.); gea@certh.gr (G.A.)
**\*** Correspondence: lmit@certh.gr

**Abstract:** The hyperloop is an innovative land transport mode for passengers and freight that travels at ultra-high speeds. Lately, different stakeholders have been engaged in the research and development of hyperloop components. The novelty of the hyperloop necessitates certain directions to be followed toward the development and testing of its technological components as well the formation of regulations and planning processes. In this paper, we conduct a comprehensive literature review of hyperloop publications to record the current state of progress of hyperloop components, including the pod, the infrastructure, and the communication system, and identify involved EU stakeholders. Blending this information results in future directions. An online search of English-based publications was performed to finally consider 107 studies on the hyperloop and identify 81 stakeholders in the EU. The analysis shows that the hyperloop-related activities are almost equally distributed between Europe (39%) and Asia (38%), and the majority of EU stakeholders are located in Spain (26%) and Germany (20%), work on the traction of the pod (37%) and the tube (28%), and study impacts including safety (35%), energy (33%), and cost (30%). Existing tube systems and testing facilities for the hyperloop lack full-scale tracks, which creates a hurdle for the testing and development of the hyperloop system. The presented analysis and findings provide a holistic assessment of the hyperloop system and its stakeholders and suggest future directions to develop a successful transport system.

**Keywords:** hyperloop; fifth mode; ultra-high speed; low-pressure; vacuum



## 1. Introduction

The hyperloop is defined as a mode of land transportation capable of high-speed and driverless operations in which a vehicle is guided through a low-pressure tube or system of tubes, for passengers and/or cargo [1]. It is a novel mode of intercity transport, designed to connect cities safely, efficiently, and sustainably, in a fixed guideway tube-based infrastructure. The hyperloop is a mode for passenger and freight transport that travels at ultra-high speeds of up to 1200 km per hour. It may also be described as a pod- and magnetic-levitation-based mode of transport in a low-pressure-sealed tube or system of tubes that operates in a low-pressure environment to reduce drag and increases efficiency to drastically reduce travel times [2,3]. A fusion of advanced technologies that is used on high-speed railway (HSR), aviation, aerospace, and magnetic levitation applications is required for the successful implementation of the hyperloop and its safe integration into the current transport system [4].

Since the first conception of the hyperloop, a significant amount of research and patent activity on several aspects of the hyperloop system components has been highlighted. Although there are other transport modes with similar components that can be used in hyperloop development, some hyperloop components differ substantively. For example, the hyperloop uses a propulsion system similar to the maglev trains but runs at higher speeds. Additionally, the pressure value that the pod supports is similar to conditions used in airplanes. Description of the hyperloop system components is provided in several

studies [5–22]. Hyperloop studies focus on specific performance topics, such as aerodynamics [15,18,22–43], safety [8,9,16,19,20,44,45], and energy [7,9,14,16–18,23,28,39,44,46–58], while others focus on different hyperloop technologies, such as in the field of pneumatic tube and tunnel systems [18,20,23,24,28,31,42,44,53,59–61]. The system functionality of certain technologies at a sub-scale level and low speeds has been proven; however, the compatibility of the various systems in subsonic speed ranges and at a real scale has yet to be verified [5,11,16,20,44].

This fusion of technologies has engaged multiple public (e.g., MIT, UPV, EPFL, ETH, TU Delft, TU Munich, etc.) and private (e.g., Hardt, Hyperloop Transportation Technologies, Virgin Hyperloop, Zeleros, Nevomo, Swisspod Technologies, Transpod, etc.) stakeholders in the research and development of the hyperloop system in North America, Asia, and Europe. Gkoumas and Christou [45] reviewed hyperloop scientific literature and patents, while the Hyperloop Standards Desk Review [3] provided a list of US stakeholders engaged in hyperloop R&D. Nonetheless, a list of EU stakeholders working on the hyperloop system and research directions per hyperloop component is not provided in the literature.

Compared to those reviews, our study adds value by reviewing the most relevant literature to (1) describe the current state of progress for hyperloop system components and clusters information, (2) identify and present EU stakeholders per country engaged in hyperloop R&D, and (3) provide directions for future research per hyperloop component.

In the remainder of this paper, Section 2 presents the methods used to identify literature for our review. The results in Section 3 include metrics and key features of reviewed publications, clustered information, identified EU stakeholders engaged in hyperloop activities, and a comprehensive analysis of the current state of progress for hyperloop components. We conclude the paper by providing directions for future research per hyperloop system component in Section 4.

## 2. Research Methodology

The hyperloop has triggered global awareness and efforts between 2015 and 2020 on improving its systems and components. A review of the existing literature is performed to answer four research questions [62]: (1) which stakeholders work on the hyperloop system, (2) where involved stakeholders are located, (3) what the current state of progress related to the hyperloop system is, and (4) what the next steps toward implementing hyperloop system are. To answer these research questions, the most recent data are collected and used. For the literature review, the data sources that are used to collect the necessary information and data include published journal and conference papers (Science Direct, Web of Science, Google Scholar, Wiley Online Library. and Springer) as well as company and project reports.

The collection of data is composed of three sub-steps: primary studies, search keywords, and search databases. Primary studies refer to the identification of relevant studies to ensure first that the set research questions–objectives are valid, avoid duplication of previous work, and ensure that enough material is available to conduct the analysis. An initial search in "Google Scholars" and "Science Direct" by using the terms "hyperloop" AND "review" resulted in two relevant studies that provide a short review of hyperloop system components [6] and a review focusing on the status of hyperloop standardization activities and stakeholder perspectives on the applicability of existing standards in the US [3]. However, none of them include an integrated review of EU stakeholders, hyperloop system components, and future directions.

The literature review focused on publications in academic journals and conferences and reports in the English language. A search was performed by using the terms "hyperloop", "tube transport", and "tube transportation". The search was performed in March 2021 and provided 195 results that included these terms in their title. The first task was to merge publications and exclude potential duplicates and publications that were not in English or related to transport, that were not accessible, or that only provided abstracts. All duplicate publications were deleted; the remaining ones were exported to an excel

file for screening. This process resulted in 107 records; 82 resulted from using the term "hyperloop" and 25 from using the term "tube transport". The following research method appears in Figure 1.

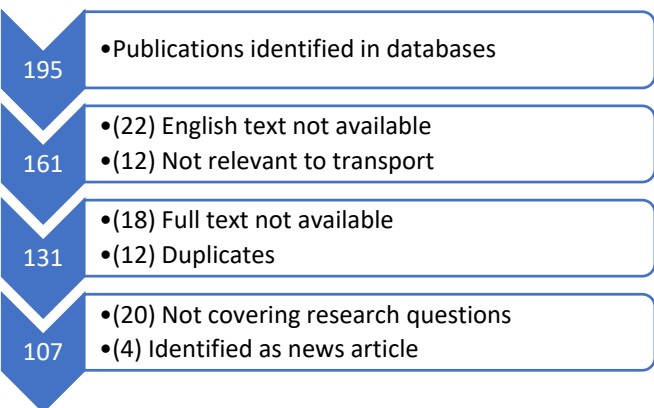

**Figure 1.** Summary of steps followed in the literature review.

### 3. Analysis and Results

The analysis started with two researchers that extracted information from publications. After the researchers reached a consensus about what to record from available publications to fulfill the study goals, they split the work equally. Firstly, a descriptive analysis of the identified literature was conducted based on the distribution of publications per country, year, and type (i.e., journal, conference, or report). Each publication was recorded according to title, authors, year of publication, and location of the study and then was reviewed to record specific features (when available) and build the database. Secondly, according to the research questions, the papers have been classified according to a) (a) stakeholder category (i.e., research organization, public organization, industry, or public–private initiative), (b) infrastructure (i.e., tube, substructure, station, interface pod–tube, or other), (c) pod (i.e., interior, system propulsion, or both), (d) performance (i.e., safety, energy, aerodynamics, traffic, environment, cost, or other), and (e) research topic. Stakeholders that were identified in publications were further explored by conducting a desk review and focusing on the online official websites of each identified stakeholder to obtain location information.

This section provides an overview of findings regarding the hyperloop publications, stakeholders in the EU, and hyperloop components, following the described method. Findings are clustered and presented in tables to better convey the information about the hyperloop's current state of progress. The goal is to gain insights related to hyperloop research and development and identify potential gaps in its operation to provide future directions.

### 3.1. Hyperloop Publications

Literature has been published from 2008–2021 (i.e., first quarter of 2021) with the majority of studies (95%) being published after 2016 (Figure 2), which shows the increasing interest in the hyperloop after the release of Hyperloop Alpha by Elon Musk in 2013 [17]. All publications before 2014 referred to "tube" transport or "vactrain".

Half of the literature at the global level is scientific journals, while the remaining 50% is almost equally distributed between conference publications and reports. In an attempt to map the geographic location of these publications on hyperloop, all authors have been recorded by country and aggregated by continent. Europe contributes significantly (39%) to hyperloop research, as shown in Figure 3. The remaining 61% is allocated to North America (24%) and Asia (37%).



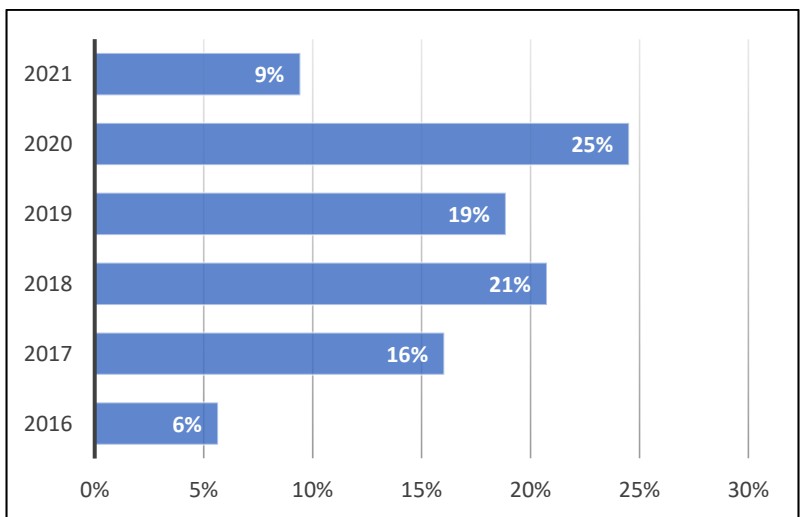

**Figure 2.** Publication year for hyperloop literature.

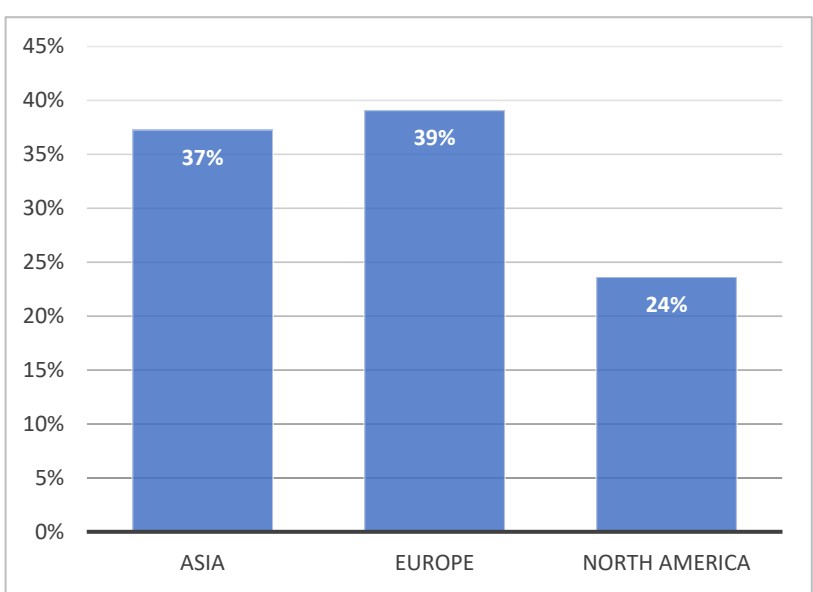

**Figure 3.** Hyperloop publications by geographic area.

Several academic and industrial research teams that focus specifically on the hyperloop system have been formed. Other transport stakeholders are also engaged in hyperloop research occasionally and publish their work on this developing field. Academic organizations usually collaborate with governmental and industry partners. The majority of the academic organizations do not focus exclusively on the hyperloop; rather, they conduct research on the hyperloop occasionally. However, some university-based teams have been developed that focus exclusively on the hyperloop. Tables 1 and 2 summarize publications in Asia and North America, respectively, in an attempt to map hyperloop activities in these two continents.

**Table 1.** Hyperloop publications in Asia.

| Source | Country | Type | Infrastructure | Pod | Performance | Research Focus |
|---|---|---|---|---|---|---|
| Bansal & Kumar, 2019 [6] | India | J | Tube | System/prop. | Other | Short review. |
| Kaushal, 2020 [12] | India | J | - | - | - | Review of the hyperloop |
| Shinde et al., 2017 [21] | India | J | Other | System/prop. | Aerodynamics | Short literature. |
| Jiqiang et al., 2020 [24] | China | J | Tube | - | Aerodynamics | Studied the differences in aerodynamic effects when the train accelerates (decelerates) past the speed of sound and the influence of different values of acceleration (deceleration) on the aerodynamic effects. |
| Oh et al., 2019 [31] | Korea, South | J | Interface pod–tube | System/prop. | Aerodynamics | Investigated the flow phenomena of a hyperloop system. |
| T. K. Kim et al., 2011 [35] | Korea, South | J | Tube | - | Aerodynamics | Studied the parameters of the tube train system: internal tube pressure, blockage ratio, and operating speed by computational analysis. |
| Zhou et al., 2021 [36] | China | J | Tube | - | Aerodynamics | Simulated the motion in the tube using the dynamic mesh method; the evacuated tube maglev train was studied under different suspension gaps. |
| Sui et al., 2020 [37] | China | J | Tube | - | Aerodynamics | Unstable aerothermal phenomenon, causing the temperature to rise sharply inside the tube and endangering the safe operation of trains and equipment. |
| Sui et al., 2021 [38] | China | J | Tube | - | Aerodynamics | Studied the influence of the vacuum degree on the flow field around a train capsule running in an evacuated tube with a circular section. |
| Niu et al., 2019 [40] | China | J | Tube | System/prop. | Aerodynamics | Formation and evolution mechanism of aerodynamic heating in the tube and influence of the Mach number at subsonic, transonic, and supersonic speed. |
| Zhou et al., 2019 [41] | China | J | Tube | - | Aerodynamics | Simulated the real motion of evacuated tube maglev train and improve the capture accuracy of the waves. |
| Tang et al., 2013 [42] | China | J | Tube | - | Aerodynamics | Study of model parameters impacting train speed and the aerodynamic drag under multifield coupling. |
| Belova & Vulf, 2016 [59] | Russia | C | Interface pod–tube | - | Energy | Analyzed the pneumatic capsule for transport of different cargoes. Studied pressure in real-time mode and movement of the capsule. |
| Le et al., 2020 [60] | Korea, South | J | Interface pod–tube | System/prop. | Aerodynamics | Investigated the effects of pod speed, blockage ratio, tube pressure, and pod length on the drag and drag coefficient. |
| Ji et al., 2018 [63] | Korea, South | J | Interface pod–tube | - | Other | Focused on thrust forces. |
| Lim et al., 2020 [64] | Korea, South | C | Interface pod–tube | - | Energy | Optimized on-board superconducting magnet with respect to energy. |
| Harish et al., 2017 [65] | India | J | Tube | System/prop. | Aerodynamics | Computational fluid dynamics (CFD) were used to simulate the airflow around the hyperloop pod at transonic speeds. |
| Rob et al., 2019 [66] | China | J | Tube | Both | - | Short review. |
| Dudnikov, 2019 [67] | Russia | C | Station | - | - | Studied the structure of the hyperloop passenger system when an intermediate station appears. |
| Dudnikov, 2018 [68] | Russia | C | - | Interior | Safety | Estimated time for the expiration of air from the capsule in an emergency situation. |
| K. K. Kim, 2018 [69] | Russia | C | Interface pod–tube | - | Aerodynamics | Alternative pipe design without using the technical vacuum. Used the rarefied air in the pipe and the linear induction motor. |

**Table 1.** *Cont.*

| Source | Country | Type | Infrastructure | Pod | Performance | Research Focus |
|---|---|---|---|---|---|---|
| S. Y. Choi et al., 2019 [70] | Korea, South | J | Interface pod–tube | System/prop. | Energy | Introduced optimal design methods for linear synchronous motors and inverters. Designed guidelines and examples for the commercialization version. |
| Dudnikov, 2017 [71] | Russia | C | Tube | System/prop. | Other | Passenger and cargo transport. Studied capacity, costs, independence from weather conditions, ecological cleanliness, and security. |
| Pradhan & Katyayan, 2018 [72] | India | C | Interface pod–tube | System/prop. | Aerodynamics | Braking forces. |
| D. W. Kim et al., 2017 [73] | Korea, South | J | Tube | - | - | CFD to investigate the shock train phenomenon inside the tube. |
| Seo et al., 2020 [74] | Korea, South | J | Interface pod–tube | - | Aerodynamics | Study of magnetic levitation driving system; design analysis. |
| Zhou & Zhang, 2020 [75] | China | J | Tube | - | Aerodynamics | High-speed movement process of evacuated tube maglev train was reproduced by numerical simulation. |
| J. Choi et al., 2016 [76] | Korea, South | J | Tube | – | Safety | Airflow through cracks. |
| Kwon et al., 2017 [77] | Korea, South | J | Tube | - | Energy | Six different photovoltaic configuration cases. |

Note: Journal (J), report (R), conference (C).

**Table 2.** Hyperloop publications in North America.

| Source | Country | Type | Infrastructure | Pod | Performance | Comments |
|---|---|---|---|---|---|---|
| NETT Council, 2021 [3] | US | R | - | - | - | A literature for domestic and international standardization activities conducted by government entities, standards development organizations (SDOs), and private industry. |
| HyperloopTT, 2019 [8] | US | R | Other | System/prop. | Other | Economic analysis and operating cost. Discussed tube, pod, vacuum system, station, and route. |
| Santangelo, 2018 [11] | US | J | Substructure | - | - | Structural approach and design. |
| MIT Hyperloop Team, 2017 [15] | US | R | Interface pod–tube | System/prop. | Other | Studied aerodynamics, energy, vibration, software. |
| AECOM, 2020 [16] | Canada | R | Other | System/prop. | Other | Tube, switching, substructure, vacuum, propulsion, and power. Energy, security, pod design, capital, and operating costs. Risk assessment. |
| SpaceX & Tesla, 2013 [17] | US | R | Other | Both | Safety | All aspects of infrastructure and pod. Route optimization. Safety, cost, and reliability. |
| Chin et al., 2015 [18] | US | J | Interface pod–tube | - | Aerodynamics | Aerodynamic and thermodynamic interactions between the two largest systems: the tube and the pod. |
| Janzen, 2017 [22] | Canada | C | Tube | - | Aerodynamics | Aerodynamics, dynamics, and vibration of tube. |
| Decker et al., 2017 [23] | US | C | Other | System/prop. | Other | Studied drag, cycle, drivetrain, geometry and mass, and levitation for pod. Studied vacuum, thermal management, propulsion, and substructure for tube. |

**Table 2.** *Cont.*

| Source | Country | Type | Infrastructure | Pod | Performance | Comments |
|---|---|---|---|---|---|---|
| Opgenoord & Caplan, 2018 [26] | US | C | - | System/prop. | Aerodynamics | Aerodynamic design considerations for the hyperloop pod (aerodynamic design considerations for the pod). |
| Covell, 2017 [46] | US | R | Other | Both | Other | Review all parts of infrastructure. Concerns and risks are outlined. Speed, time, energy, emissions, and costs. |
| Taylor et al., 2016 [53] | US | R | Station | Both | Other | Hyperloop comparisons to other modes: Travel time, frequency, user cost, comfort, reliability, energy consumption, capacity, system resilience, system interoperability, costs, and safety. |
| Bose & Viswanathan, 2021 [78] | US | J | Tube | - | Aerodynamics | Study 1) the piston effect, and 2) the addition of aerofoil-shaped fins on the performance of a hyperloop pod in a partially vacuum tunnel. |
| Sayeed et al., 2018 [79] | Canada | C | Interface pod–tube | - | Energy | A comprehensive finite-element analysis to determine the design specifications of the pod levitation and propulsion control. |
| Heaton, 2017 [80] | US | C | Substructure | - | Safety | Earthquake motion impact on tube and centripetal forces on the pod. |
| Chaidez et al., 2019 [81] | US | J | Interface pod–tube | - | Energy | Power requirements for each of the three major modes of hyperloop operation: rolling wheels, sliding air bearings, and levitating magnetic suspension systems. |
| Nikolaev et al., 2018 [82] | Canada | C | Interface pod–tube | System/prop. | Safety | Validate correctness of pod's software and embedded systems. |
| Halsmer et al., 2017 [83] | US | C | Interface pod–tube | System/prop. | Other | Develop a prototype for a high-speed, magnetically levitated transport pod for the hyperloop competition. |
| Soni et al., 2019 [84] | US | R | Interface pod–tube | - | - | Braking forces. |
| Sirohiwala et al., 2007 [85] | US | R | Other | - | Other | Maglev, high speed, cost, safety, energy, and aerodynamics. |
| Rajendran & Harper, 2020 [86] | US | J | - | - | Traffic | Built simulation models to study operational perspectives. |

Note: Journal (J), report (R), conference (C).

### 3.2. Hyperloop in the EU

This subsection presents the publications and EU stakeholders engaged in hyperloop activities. The comprehensive literature review of the hyperloop is presented in Table 3. The table presents a detailed categorization of publications per location, hyperloop components (i.e., infrastructure and pod), and performance areas. This categorization allows obtaining the required information and insights related to the hyperloop system and the components that stakeholders work on and identifying the research trends in this emerging fifth mode of transport.

It was estimated that 45% of total publications were released by a European-based organization, of which 49% were released in a scientific journal, 21% in conference proceedings, and 30% as a report. The majority of all published material (80%), including journals, conference publications, and reports, has been released by academic institutes.

**Table 3.** Hyperloop publications [1] in Europe.

| Source | Country | Type | Infrastructure | Pod | Performance | Research Focus |
|--------|---------|------|----------------|-----|-------------|----------------|
| Noland, 2021 [5] | Norway | J | Other | System/prop. | Other | A comprehensive review of the core technologies needed to realize the hyperloop transportation system, demonstrating the theoretical background. Comparison of two technical solutions and identification of future research items. |
| Janić, 2020 [7] | Netherlands | J | - | - | Environment | Energy consumption and GHG emissions. |
| Riviera, 2018 [9] | Italy | R | Other | System/prop. | Energy | Tube, substructure, and station. |
| BAK Economics AG, 2020 [10] | Switzerland | R | - | - | Other | Travel time, speed, cost, capacity, energy, environment, and safety. |
| Hansen, 2020 [13] | Netherlands | J | Station | Interior | Other | Technical feasibility of the proposed hyperloop concept for vehicle design, capacity, operations, propulsion, guidance, energy supply, traffic control, safety, alignment, and construction. Environmental impacts, investment, operating, and maintenance costs for implementation of a hyperloop line. |
| Tudor & Paolone, 2019 [14] | Switzerland | C | Interface pod–tube | System/prop. | Energy | Assessment of the optimal design of the propulsion system of an energy autonomous hyperloop capsule. |
| Delft Hyperloop, 2020 [19] | Netherlands | R | Tube | Both | Safety | Fire safety, communication system, and emergency evacuation. |
| Delft Hyperloop, 2019 [20] | Netherlands | R | Other | Both | Other | Levitation, propulsion, passenger pod, tube, vacuum, communication, artificial intelligence, cost estimation, and safety. |
| Connolly & Woodward, 2020 [25] | United Kingdom | J | Tube | System/prop. | Aerodynamics | Energy, safety, economics, and journey time. |
| Nick & Sato, 2020 [27] | Switzerland | J | Tube | System/prop. | Aerodynamics | Drag and lift forces. |
| Nowacki et al., 2019 [28] | Poland | C | Tube | - | Energy | Studied the flow of the capsule and the determination of the force acting on the nose of it. |
| Wong, 2018 [34] | Netherlands | R | Tube | - | Aerodynamics | Aerodynamic shape optimization procedure for a hyperloop pod. |
| Machaj et al., 2020 [39] | Poland | J | - | System/prop. | Aerodynamics | Aerodynamic and heat transfer study of a battery-powered vehicle moving in a vacuum tunnel. |
| Lluesma Rodriguez, González, et al., 2021 [43] | Spain | J | Tube | System/prop. | Aerodynamics | Demonstrated that the drag coefficient is almost invariant with pressure conditions. |
| Van Goeverden et al., 2018 [44] | Netherlands | J | - | - | Other | Financial, social/environmental indicators. |
| Gkoumas & Christou, 2020 [45] | Italy | J | Other | System/prop. | Other | Energy consumption, safety and serviceability, and financial feasibility. Literature review aspects. |
| Lafoz et al. [48] | Spain | J | Interface pod–tube | System/prop. | Energy | Analyzed the alternatives for the power supply of the hyperloop. Selected the technology case of the Spanish company Zeleros. |
| Pellicer Zubeldia, 2020 [51] | Spain | R | Tube | System/prop. | Other | A freight transport vehicle was conceptually developed, analyzed, and simulated. Established variables for different aspects: Kantrowitz limit, aerodynamics, transportation, energy consumption, batteries, levitation and propulsion, etc. |
| Tudor & Paolone, 2019 [54] | Switzerland | C | - | System/prop. | Energy | Optimal design of the propulsion system of an energy-autonomous hyperloop capsule. |
| Werner et al., 2016 [55] | Germany | J | - | - | Other | Speed, frequency, payload, energy, consumption, safety, traffic, noise, reliability, pollution, cost, maintenance, and shared value. |

**Table 3.** *Cont.*

| Source | Country | Type | Infrastructure | Pod | Performance | Research Focus |
|---|---|---|---|---|---|---|
| Museros et al., 2021 [61] | Spain | J | Substructure | - | Safety | The strength and stability of the tube have been analyzed by taking into account the self and dead weight, internal low pressure, wind, thermal, and traversing vehicle dynamic effects. |
| Stryhunivska et al., 2020 [87] | Poland | J | Station | - | Safety | Analysis of a designed underground station infrastructure. |
| Walker, 2018 [88] | United Kingdom | R | Other | Interior | Other | Construction tube and substructure. Performance: travel time, capacity, land implications, energy demand, costs, safety, and passenger comfort. |
| Alexander & Kashani, 2018 [89] | United Kingdom | J | Substructure | - | Other | Simulate the dynamic response of continuous bridges (safety). |
| Doppelbauer, 2013 [90] | United Kingdom | R | Other | System/prop. | Other | Summary of hyperloop system. Fundamental aspects related to innovation in infrastructure networks. |
| Munich RE, 2017 [91] | Germany | R | - | - | - | Risk assessment. |
| Ahmadi et al., 2020 [92] | United Kingdom | J | Substructure | - | Safety | Exploring the lateral dynamic interaction of the bridge deck (twin tube) and piers. |
| Voltes-Dorta & Becker, 2018 [93] | United Kingdom | J | - | - | Traffic | Planning as a complement to an airport. |
| Gkoumas & Christou, 2020 [94] | Italy | C | - | - | - | Interactions with other modes, current status in the EU, and risk assessment discussion. |
| Gkoumas & Christou, 2021 [95] | Italy | J | Other | System/prop. | Other | Safety and serviceability performance. |
| Li et al., 2019 [96] | Netherlands | C | - | Interior | Other | Embarking and disembarking process for the hyperloop. Higher efficiency and better user experience. |
| HYPED, n.d. [97] | United Kingdom | R | | - | Cost | Feasibility study, cost, and social and environmental impacts. |
| Schodl et al., 2018 [98] | Austria | C | - | | Other | Regional impacts: social, cost, and environment. |
| Munir et al., 2019 [99] | United Kingdom | R | - | System/prop. | Cost | Sustainability study. |
| González-González & Nogués, 2017 [100] | Spain | R | - | - | | Review general concept. |
| González-González & Nogués, 2017 [101] | Spain | R | - | - | Cost | Comparison of HSR, maglev, and hyperloop. |
| Connolly & Costa, 2020 [102] | United Kingdom | J | Substructure | - | Safety | Simulated ground-wave propagation in the presence of a series of discrete high-speed loads moving on a soil-guideway system. |
| Strawa & Malczyk, 2019 [103] | Poland | J | Interface pod–tube | System/prop. | Other | Performance and stability of the vehicle. Studied ride comfort of passengers traveling in a compartment. |
| Lluesma-Rodríguez, Álcantara-Ávila, et al., 2021 [104] | Spain | J | Tube | - | Aerodynamics | Used methods for extensive direct numerical simulations of passive thermal flow for several boundary conditions. |
| García-Tabarés et al., 2021 [105] | Spain | C | Interface pod–tube | System/prop. | - | Studied and compared alternatives for acceleration. |
| Vellasco et al., 2020 [106] | Spain | C | - | - | Other | Analyzed existing infrastructure network of Kazakhstan, highlighting the constraints and difficulties. Reviewed aspects of the proposed corridor from a technical, social, economic, and environmental perspective. |

[1] Note: Journal (J), report (R), conference (C).

Different EU stakeholders are engaged in activities related to the hyperloop system. These EU stakeholders refer to organizations whose main goal and/or objective is the development of the hyperloop system. To capture different aspects related to the hyperloop system, three stakeholder categories are used:

- Research and public organizations (62) (Table 4).
- Private companies (10) (Table 5).
- Public and private initiatives (9) (Table 6).

In total, 81 unique organizations have been identified in 14 EU countries. It should be noted that when more than one department of an organization is related to the hyperloop, then this organization is counted as a single entry. Figures 4 and 5 show the percentage of entries per country, with Spain, Germany, Switzerland, and the UK accounting for 26%, 20%, 11%, and 10%, respectively. All other countries are allocated a share of 9% or below.

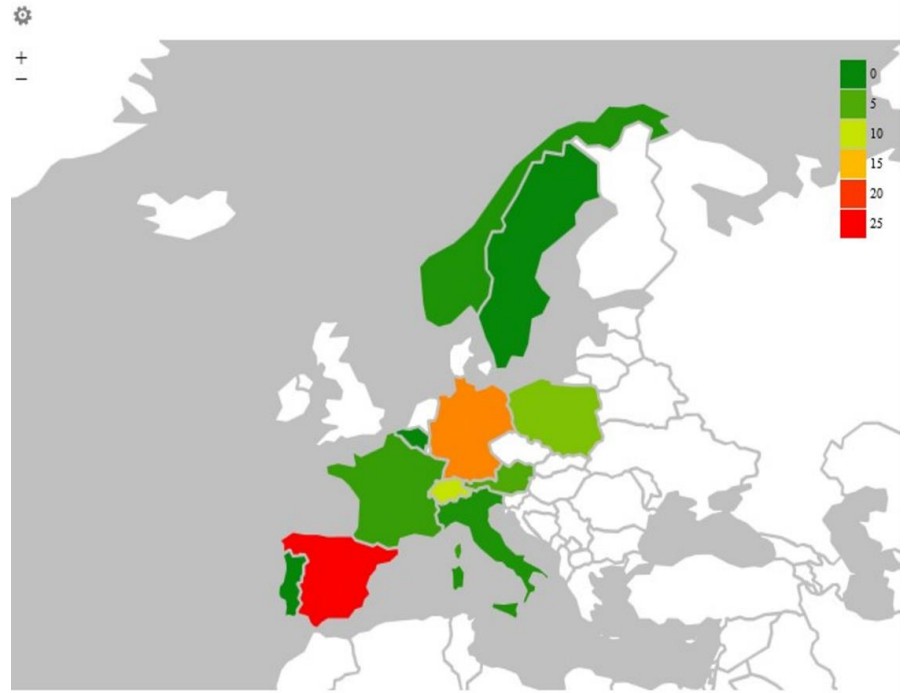

**Figure 4.** Distribution of EU stakeholders working on hyperloop.

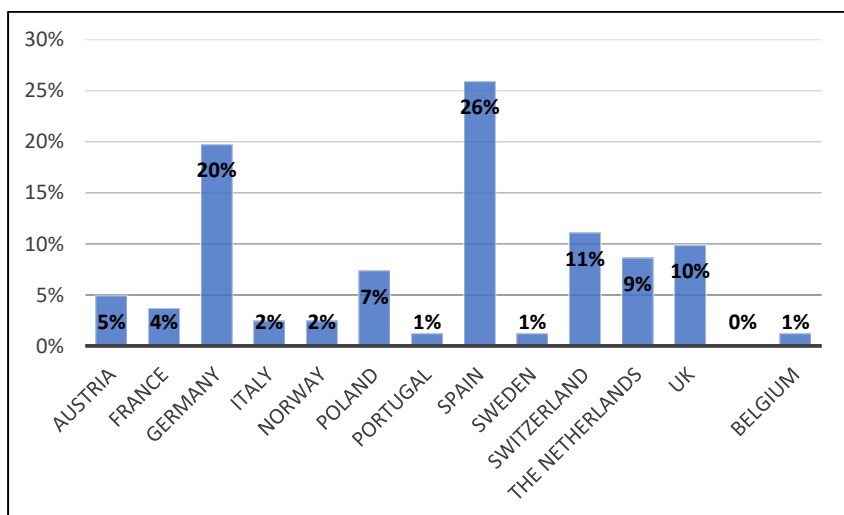

**Figure 5.** Stakeholder percentage related to hyperloop per country.

The majority of EU-based organizations that are related to hyperloop are found to be academic or research institutes (76%). In many cases, hyperloop technology developers have been actively collaborating with policymakers and Standard Developing Organizations (SDO). The stakeholder list (Tables 4–6) arranged by country may also serve as a guide for stakeholders and investors seeking partnerships for the hyperloop.

**Table 4.** Research and public organizations.

| Research and Public Organizations | Short Name | Country |
|---|---|---|
| University of Applied Sciences | UAS | Austria |
| Austrian Institute of Technology | AIT | Austria |
| University of Applied Sciences Upper Austria | FH Upper Austria | Austria |
| Vienna University of Technology | TU Wien | Austria |
| EU Agency for Railways | ERA | France |
| Deutsches Zentrum für Luft- und Raumfahrt (German Aerospace Center)—Next Generation Train | DLR-NGT | Germany |
| Hochschule für Technik Stuttgart (Stuttgart University of Applied Sciences) | HFT Stuttgart | Germany |
| Fraunhofer | - | Germany |
| Technical University of Braunschweig | TU Brauschweig | Germany |
| Technical University of Dresden | TU Dresden | Germany |
| Technical University of Darmstadt | TU Darmstadt | Germany |
| Institute of Railway and Transportation Engineering | IEV | Germany |
| University of Stuttgart | - | Germany |
| Helmut Schmidt University | - | Germany |
| Technical University of Munich (TUM) | TUM Hyperloop | Germany |
| TU Berlin | | Germany |
| The International Maglev Board | Maglevboard | Germany |
| University of Applied Sciences Emden/Leer & Carl von Ossietzky University of Oldenburg | HyperPodX | Germany |
| European Commission, Joint Research Centre (JRC) | JRC | Italy |
| Politechnico di Torino | - | Italy |
| Norwegian University of Science and Technology | - | Norway |
| Norwegian University of Science and Technology—Shift Hyperloop | Shift Hyperloop | Norway |
| Wroclaw University of Science and Technology, Faculty of Mechanical Engineering | - | Poland |
| Wroclaw University of Science and Technology, Department of Cryogenics and Aerospace Engineering | - | Poland |
| Warsaw University of Technology, Faculty of Power and Aeronautical Engineering | WUT-IAAM | Poland |
| Poznan University of Technology, Faculty of Transport Engineering | PUT | Poland |
| AGH University of Science and Technology | AGH | Poland |
| University of Porto, Faculty of Engineering | U.Porto | Portugal |
| Administrador de Infraestructuras Ferroviarias (Administrator of Railway Infrastructures) | ADIF | Spain |
| Centro de Estudios y Experimentación de Obras Publicas (Center for Studies and Experimentation of Public Works) | CEDEX | Spain |
| Centro de Investigaciones Energéticas, Medioambientales y Tecnológicas (Center for Energy, Environment and Technology) | CIEMAT | Spain |
| University of Cantabria | - | Spain |
| Hyperloop UPV—Universitat Politècnica de València (Polytechnical University of Valencia) | Hyperloop UPV | Spain |
| Universitat Politècnica de València (Polytechnical University of Valencia), Instituto Universitario de Matemática Pura y Aplicada | UPV-IUMPA | Spain |
| Universitat Politècnica de València (Polytechnical University of Valencia), Dpt. of Continuum Mechanics and Theory of Structures | UPV | Spain |
| Instituto Tecnológico de la Energía (Technological Institute of Energy) | ITE | Spain |
| University of Zaragoza, Dpt. of Industrial Engineering | - | Spain |
| Instituto Tecnológico del Embalaje, Transporte y Logística (Technological Institute of Packaging, Transport and Logistics) | ITENE | Spain |
| IKERLAN | - | Spain |

**Table 4.** *Cont.*

| Research and Public Organizations | Short Name | Country |
|---|---|---|
| Red Nacional de los Ferrocarriles Españoles (National Network of Spanish Railways) | RENFE | Spain |
| Tecnalia | - | Spain |
| Universidad Politécnica de Madrid (Technical University of Madrid) | UPM | Spain |
| Fundación Valenciaport (Valenciaport Foundation) | - | Spain |
| KTH Royal Institute of Technology | KTH Hyperloop | Sweden |
| École Polytechnique Fédérale de Lausanne (Swiss Federal Institute of Technology Lausanne) | EPFLoop | Switzerland |
| EPFL, Distributed Electrical Systems Laboratory—Power Systems group | - | Switzerland |
| Eidgenössische Technische Hochschule Zurich (Swiss Federal Institute of Technology Zurich)—Institute for Transport Planning and Systems | ETH Zurich-IVT | Switzerland |
| ETG Zurich, Department of Mechanical and Process Engineering | ETH Zurich-D-MAVT | Switzerland |
| ETH Zurich, University of Zurich (UZH), University of St. Gallen, University of Applied Sciences and Arts North-western Switzerland (FHNW) | Swissloop | Switzerland |
| Paul Scherrer Institute, Nuclear Energy and Safety Research Division | PSI-NES | Switzerland |
| The EuroTube Foundation | EuroTube | Switzerland |
| BAK Economics AG | BAK | Switzerland |
| Delft University of Technology—TU Delft | Delft Hyperloop | The Netherlands |
| Delft University of Technology, Faculty of Industrial Design Engineering | | The Netherlands |
| Transport & Planning Department, Delft University of Technology | - | The Netherlands |
| Hyperloop Edinburgh (University of Edinburgh) | HYPED | UK |
| University of Edinburgh Business School | - | UK |
| University of Strathclyde | Stathloop | UK |
| University of Sheffield, Department of Mechanical Engineering | - | UK |
| University of Leeds, School of Civil Engineering, Institute for High-Speed Rail and System Integration | IHSRSI | UK |
| University of Southampton | - | UK |
| University of Bristol | - | UK |

**Table 5.** Private companies.

| Private Companies | Short Name | Country |
|---|---|---|
| TransPod | - | Canada, France |
| IKOS Consulting | - | France |
| Munich Re | - | Germany |
| Nevomo | - | Poland |
| Zeleros | - | Spain |
| ROADIS Transportation Holding | Roadis | Spain |
| Swisspod Technologies | Swisspod | Switzerland |
| Hardt | - | The Netherlands |
| TRL | - | UK |

In terms of EU publications, almost half of the publications are journals (49%), and 21% are conference publications, showing the academic interest in the hyperloop system and the increasing research attempts in different aspects of the system. The remaining 30% is allocated to reports, which are associated mostly with private-based stakeholders.

An analysis of the published literature is performed on the basis of infrastructure, pod, and performance to gain a deeper insight into the hyperloop components and performance goals that stakeholders work on. Infrastructure is divided into the (1) tube, (2) substructure, (3) interface pod–tube, (4) station, and (5) other. "Other" covers publications that are not covered by the four identified infrastructure areas or that refer to generic infrastructure aspects. The pod is divided into the (1) system and propulsion, (2) interior, or (3) both. Finally, the performance of the hyperloop system is explored by considering six areas:

(1) safety, (2) energy, (3) aerodynamics, (4) traffic and capacity, (5) environment, (6) cost, and (7) other. "Other" refers to publications that include two or more areas or areas that are not covered within the six areas. Publications that focus on legislation were considered separately; a necessary area of research for the development of the hyperloop. Figure 6 summarizes the results of this analysis. It is noted that one publication may be attributed to one or more areas; therefore, the total sum may not be 100%.

**Table 6.** Public and private initiatives.

| Private and Public Initiatives | Short Name | Country |
|---|---|---|
| CEN-CLC/JTC 20—Hyperloop systems | - | Belgium |
| EU HyTeC | - | Germany |
| Institute of Hyperloop Technology | IHT | Germany |
| European Hyperloop Week | EHW | Spain |
| European Hyperloop Development | - | Spain |
| MAFEX's Hyperloop—Hyperloop Spanish Observatory | | Spain |
| Railway Innovation Hub's Hyperloop Strategic Working Group | RIH | Spain |
| European Hyperloop Program | - | The Netherlands |
| European Hyperloop Center | EHC | The Netherlands |
| Hyperloop Connected | - | The Netherlands |

Hyperloop studies are found to conduct research related to the traction of the pod (37%) within the tube (28%) and quantify impacts related to safety (35%), energy (33%), and cost (30%). Other hyperloop areas, including passenger comfort and system acceptance, while focusing on substructure and station, are scarcer in the EU literature. For the organizations that focus on the hyperloop pod, the majority of them focus on the exterior design (i.e., related to aerodynamics), whereas only two studies were found to focus specifically on the interior. Other fields of research relate to social impacts, land implications, serviceability, and hyperloop maintenance.

Different fields of research are engaged in the hyperloop system, including mechanical, transport, electric and aeronautical engineering, and business and structural experts. In terms of transport modes (when such information was available), 7% of EU entities relate to aviation, 44% relate to high-speed rail, 25% relate exclusively to the hyperloop, and 24% relate to road transport.

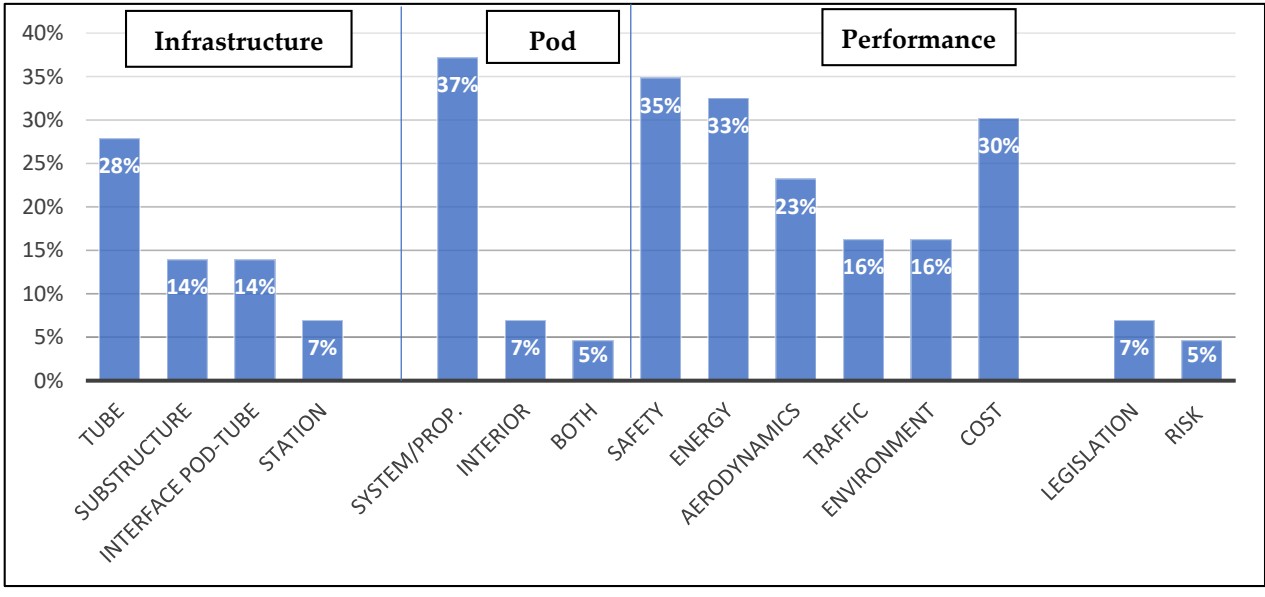

**Figure 6.** Hyperloop EU research areas.

### 3.3. State of Progress for the Hyperloop System

Based on the literature findings, the current state of progress for the hyperloop system is provided by focusing on the components of the system. These are:

- The vehicle (also called pod or capsule), which includes an aerodynamic fuselage (similar to the construction of a commercial aircraft), the interior, and the electric subsystem.
- The infrastructure, which includes the tube, the sub-structure, and the stations. The tube encloses and maintains the low-pressure environment, ensuring minimum air leakage. The infrastructure also contains the pressure maintenance system and the power substations, which provide a considerable reduction of air drag, allowing for a smooth travel of the pod with speeds of up to 1200 km/h. Infrastructure characteristics depend on the type of levitation and propulsion system.
- The communication system, which creates an autonomous environment, exchanges data, and coordinates operations, ensuring safety and comfort.

### 3.4. The Pod

#### 3.4.1. Structure

The pod is the main structural frame of the system and is considered equivalent to an aircraft airframe [15,20]. The hyperloop pod is effectively a pressure vessel to withstand pressure differences and, most importantly, to transport people and cargo [9,18–20,107]. Its design combines aerodynamics, materials technology, and manufacturing methods with a focus on performance, reliability, and cost [16]. The pod is designed to be light as possible to accommodate external low-pressure conditions, design speed, and include on-board systems and interior furnishings, maximizing passenger safety, travel experience, and comfort, inside a tube-based environment [8,20,90,107]. The pod design significantly affects the design of the tube infrastructure, which depends on the loading pad configuration and the formation of distributed or concentrated loads [11], as well as the static, dynamic, and thermal loads [61]. The frontal surface and shape of the pod may affect the aerodynamic drag and the tube's operational energy consumption [5,14,28,31,36,39,51]. The Kantrowitz limit, the blockage ratio, the drag coefficient, and the pod's length are certain factors for optimizing the aerodynamic performance and pod speeds [5,18,22,24,26,27,34,41,43,51,60].

#### 3.4.2. Pod Interior

Passenger safety and comfort inside a pod are based on a combination of best practices from rail and aviation transport, containing certified components of mature technologies [8]. A human-centric interior design with augmented reality windows [17], lighting, bright colors [19], texture, and control of sound levels will provide comfort, journey information (i.e., time to destination, exact location, speed, time, simulations/videos of the external environment), and entertainment to the passengers [108]. Interior furnishings and different evacuation options for emergency cases within the pod will be considered to maximize passenger safety and travel experience [16,19]. Moreover, pod interiors will be designed to include first aid kits [16,17,19,53], automated external defibrillator machines, and an emergency response call/communication system [16]. Conceptual designs of well-supported seats with seatbelts to protect passengers from rapid acceleration and deceleration have been demonstrated; however, testing seat design and safety aspects for passengers at high operating speeds is required to establish their viability [16,19,20].

#### 3.4.3. Power System

Two concepts of power systems are identified [5,16,109]: (1) The first uses a guideway as a propulsor, creating a lightweight pod that needs a pricey infrastructure cost, and (2) the second uses an energy-autonomous pod by storing massive amounts of onboard energy, thus significantly lowering the infrastructure costs.

- On-board rechargeable batteries may provide power supply to the pod's system [16]. Each pod may supply its own power for levitation, acceleration, control, and other amenities. Fast-charging systems that store regenerative breaking energy are under

development [110]. Challenges include thermal management, power requirements, infrastructure cost, availability of pods, and charging intervals [16].

- An infrastructure-side power system, connected to the electrical grid, may demonstrate several advantages, including better efficiency, reduced energy consumption, potential cost savings on the construction of the tube, higher manufacturing tolerances of the guideway, and synchronized control of propulsion, decreasing the potential of collisions. However, high infrastructure costs, fault tolerance, and the acquisition of considerable land area to house the electric substations are certain limitations associated with this system [16].

*3.5. Infrastructure*

### 3.5.1. Tube

The tube provides a low-pressure travel-guideway environment and protects the pod from all external conditions [20]. It is airtight to maintain the low-pressure environment [10,11,16,19,20,85,95,106,111], strong enough to prevent failures [11,16,17,19,20,25,61,80,89,102] and designed according to the geometry of the pod and the aerodynamic requirements [16,18–20,23,27,31,34,42,44,59,61,71]. Tube geometry depends on the operational pressure level of the system [18,20,31,34,42,60,61,109]. Additionally, the tube needs to be grade-separated from other transport modes [8,16,44,112]. Concrete pylons are expected to support the tube system with a height depending on the configuration of the guideway [53,107]. Three types of guideway infrastructures are under study: elevated, on ground, and underground [8,9,11,13,16,22,66,87,107]. The elevated guideway is expected to be the safer, as there is no need for crossing control systems at roadway intersections [107] and the land footprint is smaller for pylons compared to a railway track [44,107]. Leveraging the surface on top of the tubes, solar panels could be installed [8–13,17–21,25,44,53,66,87,88,106] depending on local conditions, thus contributing to the local electric grid.

### 3.5.2. High-Speed Switches

The switches are track-changing mechanisms, which allow pods to pass from one track to the other, realizing a point-to-point connection in a network of tubes and connecting various cities [19,20]. For switches, two primary scenarios have been envisioned [16]. The first envisages allowing tubes to diverge toward different destinations. The second splits the main tube in two, several kilometers away from a terminal, providing additional capacity for the acceleration and deceleration stages. A larger number of portals (i.e., terminal entry or exit points) will be needed if actual pod volumes approximate forecasts [16]. The capability of the pods to switch between tubes is expected to be enhanced by the development of switching technology [16,20]. The hyperloop system cost is highly dependent on the number of required switches [16,20]. Maintenance and monitoring of the high-speed switches are also required [16,20] to ensure lateral guidance and safety, avoiding unexpected collisions [20]. The only switch technology demonstrated publicly, at low speeds and at a reduced scale, is claimed to be at Technology Readiness Levels (TRL) Level 4. However, as there are plenty of challenges for a magnetic-based switching system, this technology is estimated to be at TRL Level 2 [16].

### 3.5.3. Airlocks

The airlocks are devices equipped with gate valves that allow the loading and unloading of hyperloop pods inside the evacuated tube, without re-pressurizing the whole tube, thus facilitating the transition from atmospheric pressure to low pressure and vice versa [10,19,20,90]. When the pod frequency is high, an arrangement of multiple parallel operating airlocks is required to increase the speed and efficiency of (dis)embarking [16]. Airlock development can be divided into two main concepts [113]:

- An airlock chamber in which pressure variation is expected. The airlock chamber acts as a pressure regulator, ensuring the transition from atmospheric pressure to a low-pressure-tube environment and the reverse; depressurization occurs once the pod is sealed.
- Bridge doors at the platform that will lock onto the pod doors allowing the pod to be exposed only to a low-pressure environment and connecting the pod to the station atmosphere.

Two options for locating the airlocks are researched: placing the airlock chambers in the station hall or in the low-pressure zone [87]. In terms of safety, operation time, and minimum required area for (dis)embarking, the first concept is the most viable [113].

### 3.5.4. Pressure Maintenance System (PMS)

The PMS is responsible for the initial evacuation of air (pump down) and the maintenance of the steady-state conditions, as well as managing air leakages [16,19,20,23]. PMS may rely on different pressure levels, with some of them working at a pressure similar to civil aviation and others similar to space [109]. Defining an optimum pressure level is a trade-off between the required power to maintain the pressure and the power to overcome the aerodynamic drag [20,23,109]. The power requirements to depressurize the tube on time and maintain the low-pressure environment in the tube depend on the tube size [18]. A combination of backing pumps to create a low-pressure environment and root pumps to reach and maintain the pressure to required levels is used to overcome the aerodynamic drag [20]. The energy consumption for the pressure maintenance system is significantly high [9,23], while the initial pump-down is the most energy-consuming process that results in a high cost [16].

### 3.5.5. Interfaces—Levitation

The first hyperloop concept proposed the use of air-bearings for levitation combined with a linear induction motor [17], which requires high maintenance, tight integration between the track and the pod, and significantly increasing the pod weight with the use of fans, motors, and hover-pads [15,20]. Subsequent efforts focused on magnetic levitation (maglev) that can be coupled with electromagnetic propulsion for higher efficiency [23]. According to recent levitation comparison studies [5,20], the most dominant technology for ultra-high speeds is the electromagnetic suspension (EMS) and the electrodynamic suspension (EDS), called "active levitation" and "passive levitation", respectively.

- The EMS technology is based on the attractive properties of the magnets, and it uses pod-side electromagnets and ferromagnetic materials on the guideway [5,7,16,20,52]. The EMS works with magnetic forces giving a lower levitation of about 10–20 mm above the guideway [11]. Optimization solutions have been recently reported using hybrid options (H-EMS) [114,115] or high-temperature superconductivity (HTS-EMS) [116,117].
- The EDS technology is based on electromagnetic induction, and it uses pod-side permanent magnets or superconducting electromagnets and a highly conductive guideway infrastructure that generate opposing magnetic fields through induction [5,7,16,20,52]. In EDS, the pod can be levitated 10–100 mm, using permanent magnets (PM-EDS) or superconducting magnets (SC-EDS) [11]. Another technology that uses embedded conducting wire loops to minimize the eddy current from the moving magnetic array is the Inductrack [118,119]. Using high-temperature superconductivity for EDS (HTS-EDS), it supports speeds of up to 620 km/h [120]. It should be noted that a hybrid solution using both EMS and EDS has also been proposed [121].

### 3.5.6. Interfaces—Propulsion

The main functions of the propulsion subsystem are to accelerate the pod, to have the ability to brake or decelerate, to sustain the target speed between the acceleration and deceleration zones, combatting drag forces, and to provide safe magnetic field levels and comfort in the passenger compartment [5]. Currently, two types of propulsion systems are under consideration: the axial compressors and the linear motors [16]. The latter include

the linear induction motor (LIM), the linear synchronous motor (LSM), and the linear switched reluctance motor (LSRM).

- The axial compressors compress air in front of the pod and generate thrust by forcing it out of the back at higher energy. At low speeds, compressors could be considered impractical for the initial acceleration phase; however, air can be accelerated to generate thrust if a small bypass or a large blockage ratio is considered [18].
- The LIM is a rotary motor consisting of a stator, which generates a varying magnetic field across an air gap and a rotor [20]. A series of magnetically conductive poles are arranged in a linear sequence, and time-varying electric currents are driven through windings on the poles to create time-varying magnetic fields interacting between the tube and the pod.
- The LSM consists of a number of permanent magnets on the pod and coils on the guideway, creating a traveling magnetic field [122]. Specifically, a stator that is located beneath the guideway produces a magnetic field along the guideway, and an excitation system located onboard the pod stimulates the levitation electromagnet to produce an excitation magnetic field [11]. The LSM is able to achieve high pod speeds, and it may be a reasonable choice for an energy-efficient hyperloop.
- The LSRM contains ferromagnetic poles and a secondary ferromagnetic part, separated from the primary by an air gap. Three configurations of LSRM were compared: a single-side horizontal LSRM, an N-side vertical LSRM, and a cylindrical LSRM [105]. The LSRM has been proposed for a wide range of applications, which include autonomous railway vehicles, due to its potential for high energy efficiency [123].

### 3.6. Communication System

Several types of communication systems are required, including (1) the communication of the pod's sensor data to and from a centralized data processor and (2) the communication related to the pod's location between the pod and the tube [20]. Certain challenges exist related to the pod communication and the collection of data, as well as the high-speed connection between the pods and the infrastructure [16,20,94,124–128].

A new generation of optical fibers allows the wireless communication between antennas with radio waves, [9,10,19,20]. The GSM-R (Global System for Mobile Communications-Railway) is the primary communication system [20] for HSR; however, due to certain limitations [128,129], the LTE-R (Long-Term Evolution-Railway) may be used in communication systems. The LTE-R provides capabilities for data transmission, Internet access, and high-quality voice or mobile video broadcasting [130]. Innovative systems have been also developed, such as radio and fiber networks, with dedicated antennas placed at intervals along the hyperloop system and hardware installed on the pod using the latest 802.11 Wi-Fi standards [131].

### 3.7. Hyperloop Test Tracks

An overview of the current status of the development of pods, tube systems, and testing facilities for the hyperloop is provided to complete the identification of active actions in terms of available infrastructure and test locations. Hyperloop companies and organizations, which are involved in the development of pods and tube systems for the commercialization of hyperloop, are mentioned below as "developers". A number of privately funded companies and public institutions have already built facilities, aiming to perform full-scale tests. However, only one developer (i.e., Virgin Hyperloop One) has introduced pod and tube infrastructure in full-scale, while in November 2021, they tested the world's first passenger journey. Table 7 summarizes the current status of progress of different developers and the characteristics of the testing track facilities (i.e., scale, technology, maximum reported pod speed, etc.) at a global level.

**Table 7.** Current status of the development of hyperloop testing track facilities in different scales.

| Name | Location | Max. Tested Speed (km/h) | Propulsion | Levitation | Testing Track Facilities | Scale | Reference |
|---|---|---|---|---|---|---|---|
| Hardt | The Netherlands | - | LSM | EMS | Completed Length 30 m, diameter 3.2 m | Full scale 1:1 | [20,132,133] |
| Hyperloop TT | France, US, UAE, Germany | - | LSM | EDS | France: Completed length 320 m, diameter 4 m. UAE: Developing a 4.8 km passenger track and test track 1 km US: multiple routes under study Germany: 100 m cargo route | Full scale 1:01 | [8,134,135] |
| KRRI | South Korea | 1019 | LSM | EDS | 60 m track, 20–30cm vehicles | 1:17 | [136–138] |
| Nevomo | Poland | 50 | LSM | EDS | Completed—length: 48 m Developing—length 500 m | 1:5, Planned tests at 1:1 | [139,140] |
| SwissPod | Switzerland | - | LIM | EMS | Developing—40 m length | Unknown | [16,141,142] |
| Transpod | France, Canada | - | LIM | EMS | Developing at 3 km, diameter 2 m | 1:4, Planned tests at 1:2 and full scale 1:1 | [143,144] |
| Virgin Hyperloop One | USA, Saudi Arabia, India | 387—Devloop 173 passenger test | LIM | EDS | Completed in the US—length 500 m, diameter 3.3 m. Developing full-scale projects in the US, Saudi Arabia and India | Full scale 1:1 | [145,146] |
| Zeleros | Spain | - | Compressed air Electric–aerodynamic | EMS | Completed: Six key subsystem prototypes. Developing 3–4 km tube test-track for system integration at high speeds. 20–40 km track for commercial certification and manned tests by 2030. | 1:3, Planned Tests at 1:1 | [147,148] |

## 4. Discussion

The hyperloop development is in the preliminary stages, and improving the technology readiness levels depends on initiatives and collaborations with both the private and public sectors. Understanding the current status and goals that are set for the hyperloop development results in directions that should be followed to bridge potential gaps. The following subsections use the presented hyperloop system categorization and blend literature findings to provide future directions toward developing a successful hyperloop system.

### 4.1. Future Directions

#### 4.1.1. Pod and Tube Design

The pod frontal surface and shape affect the aerodynamic drag and the tube's operational energy consumption; therefore, factors such as the Kantrowitz limit and the drag coefficient should be further researched. The optimum blockage ratio, the aerodynamic performance, and the material characteristics have the potential to contribute to lighter pods and reduced aerodynamic drag. Certain limitations still exist regarding the tube design, including lack of real-scale test facilities, standardized dimensions for tube diameter, as well as materials and proof of concept for dimensional stability. Given that very long tube lengths may experience thermal expansion variance of up to 300 m, the installation of thermal joints and connections for different tube joints is a complex task that should be addressed. System simulations are required to define the optimum aerodynamic pod with adequate passenger-carrying capacity, capable of reaching the anticipated ultra-high speeds. The tube prototypes should be tested for leakage rates at lower speeds and full-scale conditions to verify the exact diameter and dimensional stability.

### 4.1.2. Pod Interior

Conceptual designs of well-supported seats with seatbelts to protect passengers from rapid acceleration and deceleration have already been demonstrated; however, testing seat design and safety aspects for passengers at high operating speeds is still required to establish the viability of the concept. Passenger comfort needs to be satisfied for vehicle acceleration/deceleration in curves and switches, and the optimal seating arrangement should be verified to ensure passengers' safety. Traveling at ultra-high speeds is an innovative concept that is offered by the hyperloop; therefore, the sustainability of the hyperloop depends also on the passengers' acceptance of such a system. Measuring the noise level within the cabin and assessing passengers' comfort when traveling with simulated windows at ultra-high speeds is still an unexplored field of research that future attempts should focus on. Studying the pod's comfort by simulating multiple aspects, including seat comfort, thermal comfort, crowdedness, psychological distress, noise, motion sickness, and access to facilities, becomes essential for an optimum interior design.

### 4.1.3. Airlocks and High-Speed Switches

Commercial airlocks are still untested components. The number and characteristics of airlocks should be defined to ensure fast, efficient, and friendly passenger boarding/disembarking, while airlock maintenance and monitoring plans should be developed. The airlock deployment is affected by the need to (1) maintain a constant low-pressure over long tube segments, (2) eliminate pressure failures, and (3) create a communication system for managing the system. The switching process would occur at operational speeds of roughly 600 km/h; however, such a concept should be evaluated through simulations and in a real-scale environment. Understanding the possible failures and the risks involved in changing tube guideways at ultra-high speeds will enhance the safety of such state-of-the-art technology.

### 4.1.4. Pressure Maintenance System

The initial pump-down operation should be evaluated in terms of energy consumption, cost, time, maintenance, and monitoring to ensure low-pressure conditions over long segments. An in-depth analysis of the pressure pump system combined with computational fluid dynamics simulations would significantly enhance the definition of the exact tube pressure value (s).

### 4.1.5. Propulsion

The short-term goal related to the propulsion system is to assess two eligible designs: integration inside the pod or along the various segments of the infrastructure. The assessment should include aspects related to power consumption, cost, reliability, safety, monitoring, and maintenance. If linear motors will be deployed along the tube, the infrastructure cost increases significantly. In this case, linear motors should be tested for maximum acceleration and deceleration. Additionally, the pod weight increases due to on-board power supply or contactless power transfer systems to sustain the electric propulsion. To decrease infrastructure costs, axial compressors could be used. Moreover, the use of two or more systems for propulsion/levitation and guidance may complicate the system and increase maintenance costs. To that extent, several concepts have been demonstrated using the non-symmetric double-sided linear induction motor (NS-DSLIM) [63,74], performing the aforementioned functions in one system. However, finite element analysis (FEA) simulations are required to optimize the ratio of levitation to thrust force. R&D should focus on experimental demonstrations of different concepts to evaluate the performance and efficiency of the propulsion systems at ultra-high speeds and longer tracks.

### 4.1.6. Levitation

The magnetic levitation evaluation should assess the trade-offs between energy consumption, infrastructure costs, and operation reliability. For ultra-high speeds, the feasi-

bility of EDS or EMS still needs to be proven. Furthermore, a cost comparison is required to assess the infrastructure interface, since the electro-dynamic suspension (EDS) system requires a conductive material, and the electromagnetic suspension (EMS) requires an electrical steel rail. For the EMS, the active control system should be optimized to prevent failures of sensing or malfunctions. Operation stability and comfort should be also ensured against dynamic vibrations caused by guideway irregularities. Studies should be conducted to evaluate a guideway configuration for low energy consumption, low lift-to-drag ratio, and energy efficiency of the Hybrid-EMS, taking into account the power losses as a major concern for hyperloop speeds. Optimization techniques such as genetic algorithms and magnetic system analysis using numerical methods (FEM) [115] could be used to optimize the Hybrid-EMS system.

Regarding the Inductrack, studies evaluating the feasibility of permanent magnets, taking into account maintenance and operational costs, are very important to demonstrate a highly energy-efficient solution at a low cost. Finally, the performance of high temperature superconducting magnetic suspension (HTS) for both EMS and EDS should be evaluated by comparing power losses, compatibility to high speeds, reliability, and stability, according to various HTS material configurations.

### 4.1.7. Power System

The propulsion system influences the compatibility of the levitation system and impacts the overall performance and lifecycle costs. On-board rechargeable batteries may provide power supply to the pod's system and multiple options to achieve fast charging are under development. Certain challenges of using battery systems should also be solved, including thermal management, mass saving considerations, power requirements, infrastructure cost, availability of pods, and charging intervals. Apart from the use of batteries, supercapacitors may be considered for power supply, due to their power/energy ratio and cycle capability [48]; however, their adoption is prevented in high power applications. Moreover, the use of hydrogen fuel cells could contribute to reducing the pod weight even more and provide a solution to cool down the system [85,148,149].

Concerning the pressure maintenance system, for a given leakage rate, a trade-off analysis should be conducted. Lower pressure increases the power demand in the vacuum system to pump the tube down and maintain tube pressure, while higher pressure increases the power requirements [23]. The initial pump-down is considered to be the most energy-consuming process with a large impact on cost [16]. According to many design proposals, power could be delivered through renewable energy sources; however, no details have been provided, and their feasibility has not yet been proven. The study of renewable resources (i.e., solar panels) is highly important in order to solve electrification issues and to utilize the land occupied in a more sustainable way.

### 4.1.8. Communication System

Creating a communication system capable of operating with high capacity within an automated system at ultra-high speeds and a tube-based environment is very challenging. Optical fiber and wireless communication between antennas with radio waves is considered an effective option for data transmission and should be further investigated. Due to certain limitations of the GSM-R and the rapid growth of railway services, the LTE-R (Long-Term Evolution-Railway) shall be considered the next generation communication system, providing capabilities for data transmission and passenger services such as Internet access and high-quality voice or mobile video broadcasting [130]. However, due to speed limitations of 500 km/h, the current railway communication system, based on LTE, is not compatible with the expected hyperloop speeds. A leaky waveguide solution with a centralized, cooperative, Cloud Radio Access Network (C-RAN) has shown promising results for the bidirectional communication link between the train and the ground [127]. New generation 802.11 networks (WiFi 6) and 5G NR could also be considered as an option; however, network configuration and optimization are required for further evaluation [124].

4.1.9. Real Scale Test-Tracks

Real scale test-tracks with minimum dimension requirements are essential for the development and testing of the hyperloop system. In Europe, there is a strong public–private collaboration for the development of testing facilities that will enhance the research and development (R&D) of sustainable vacuum transport technologies. Two testing infrastructures are developed in Europe:

- The EuroTube [150], a non-profit framework based in Switzerland. In 2022, it plans to create the world's first publicly accessible vacuum transport test track at scale 1:2, 3.1 km in length, and 2.2 m in diameter, enabling the testing of cargo pods at speeds between 700–900 km/h. An approximately 30 km long test track is under study for development in 2026.
- The European Hyperloop Center (EHC) plans to create in 2022 a testing facility that will house a 2.6 km long track with a cargo scale tube of 1.4 m in diameter [151].

4.1.10. Transport Engineering

The hyperloop introduces new attributes to the design of the infrastructure. The speed, which is the main component that differs from traditional transport modes, affects alignment parameters including the cant, the horizontal curvature, and acceptable rates of change for vertical and horizontal cant and jerk. Additionally, in the design of vertical curvatures, the utilization of vertical transition curves (e.g., clothoid) may be explored to ease the dynamic effects on human bodies that are caused by hyperloop acceleration and deceleration. Only a couple of research publications in the literature refer to alignment aspects and discuss potential changes in the tube alignment. There is an emerging need to expand the current state of practice for designing HSR and maglev rail alignments to include hyperloop requirements. The hyperloop guideway is directly affected by the applied alignment. The guideway may be elevated, on-ground, and/or underground. On-ground and elevated guideway structures may be preferred in the shor term, due to cost advantages; however, in the long term, underwater tunnels capable of supporting a hyperloop system may provide an alternative to aviation. Conduction of simulations by considering the aforementioned changes and drafting hyperloop design guidelines for transportation engineers are essential elements for hyperloop realization.

4.1.11. Transportation Planning

Similar to design, the hyperloop, due to its innovative characteristics, both operational and physical, introduces new aspects in transportation planning. A demand forecast method needs to be established to provide rational assessments with other competing modes in different corridors. Surveys (e.g., stated preference) may also be used to collect data for potential hyperloop passengers, thus identifying their reaction to the introduction of the hyperloop. Time and cost are two major parameters in the evaluation of the hyperloop. Time is inversely proportional to speed; thus, ticket costs should be explored as a function of infrastructure cost and the passengers' value of time. Users' perception of the value of travel time (VTT) and behavioral aspects should be also captured by surveys at the national and international levels. The physical characteristics of the hyperloop (e.g., tube, vacuum system, etc.) will affect the station design (i.e., number of tracks, platform sizes, passenger, and freight waiting areas), the station type (under/overground), and the layout of the connecting facilities (i.e., hubs) with other transport modes. Developing a simulation software for a hyperloop system that embraces the aforementioned issues will be necessary to provide robust and reliable assessments.

Generally, the review reports substantial stakeholder interest in the development and testing of hyperloop components in a simulated environment or reduced-scale conditions. Performance is assessed mainly by considering safety, energy, and cost aspects. This focus is probably influenced by stakeholders' need to demonstrate the feasibility of the hyperloop technology and assess its performance related to other transport modes. However, the literature showed that little effort has been put into legal and risk aspects, with only 12%

of EU studies exploring related issues. Currently, the European legislation is divided into four main areas: road, rail, waterborne, and aviation. Hyperloop is an innovative mobility solution, partially integrating subsystems from rail and aviation, thus requiring the formation of customized regulations to support its development and operation.

Our study suffers from certain limitations. The exclusion of theses and dissertations could have been a limiting factor in that it is possible that new findings might have been overlooked. Similarly, the inclusion of company reports might have led to some generalities, although the authors attempted to reduce general remarks by integrating scientific papers and findings. Hyperloop was considered to be a ground transport mode in this study, which led to the exclusion of information that is related to underwater development and operation of the hyperloop system. Additionally, the number of EU stakeholders that was identified might have led to an insufficient interpretation of the current situation, thus omitting the global perspective. Finally, the review has been conducted in the English language, missing foreign governmental documents that may contain more information, especially on legal and legislative aspects. These limitations suggest that it would be valuable to consider foreign governmental documents and conduct a similar review by considering North American and Asian stakeholders.

## 5. Conclusions

The development of the hyperloop system attracts interest from private and public stakeholders, industrial leaders, and R&D entities. Since the release of the first hyperloop study [17] in 2013, a remarkable and rapid evolution of the hyperloop technologies has been observed. This study conducted a comprehensive literature review of the hyperloop system to explore the current state of progress of hyperloop components, identify related stakeholders, their geographic location, and their work object, and provide future directions to bridge existing gaps.

The analysis showed that Europe and Asia demonstrate particular interest in the hyperloop system. More specifically, 81 unique EU stakeholders were identified to work on pod or infrastructure components; however, very few of them were found to work on pod interior or user acceptance. The majority of the publications were found to focus on performance aspects, such as safety (35%), energy (33%), and cost (30%), while only a couple of studies focused on legislation and risk analysis.

Stakeholders aim for hyperloop commercialization and solving technical issues that arise from testing at a real scale. The design of the hyperloop's technological components is in progress, and developers appear to focus on power system requirements. Although the hyperloop is foreseen to be powered by electricity and renewable energy sources, there are no details about the feasibility of the power delivery method. Establishing the technical requirements, creating a heat management plan, and conducting a feasibility analysis of the evaluation of various concepts for power delivery are certain considerations to be solved in the long term. The system functionality of certain technologies at a sub-scale level and low speeds has been proven; however, the compatibility of the various systems in subsonic and transonic speed ranges and real scale has to be verified. For the operational speeds of a hyperloop, the feasibility of both EDS and EMS technologies still needs to be proven. The comparison study between the two led to the conclusion that the EMS demonstrates a better potential against the EDS due to its lower power consumption. Constant power supply is required for both systems; however, the energy consumption using permanent magnets can be reduced significantly by developing a suitable Inductrack guideway.

Although these aforementioned findings aim to contribute toward hyperloop development in Europe, certain impediments to its progress exist. These include, but are not limited to, the design of its sub-components, which are under study, and the operational testing at real-scale and ultra-high speeds, which is under consideration. Finally, transport engineering and planning guidance that embrace the hyperloop's physical and operational characteristics should be developed for the hyperloop design and implementation.

The review may serve as a guide for stakeholders and investors seeking partnerships and establishing research collaborations for the hyperloop, while the proposed research directions aim to provide a pathway of actions capable of bridging the identified gaps in hyperloop development. Our findings provide solid insights into the current state of progress of the hyperloop system that may be used by interested stakeholders to extend their R&D activities and identify potential gaps that need to be addressed for developing the hyperloop.

Future attempts may be focused on further expanding the stakeholder list at the global level and gradually disaggregating hyperloop components to expose the system complexity and engage more stakeholders in R&D activities. Then, a comprehensive database may be built to link stakeholders and hyperloop components to facilitate collaborations and enable knowledge sharing. In terms of system performance, each hyperloop component may be explicitly studied to assess safety, energy, and cost impacts; performing a detailed study and understanding potential tradeoffs will enhance decision-making. Finally given the innovative nature of the hyperloop, an update of the current state of progress, for both technology and stakeholders, will be required on a regular basis for the development of the hyperloop system.

**Author Contributions:** Conceptualization, L.M. and A.K. (Annie Kortsari); methodology, L.M.; validation, A.K. (Annie Kortsari); formal analysis, L.M.; investigation, L.M. and A.K. (Annie Kortsari); resources, L.M.; writing—original draft preparation, L.M. and A.K. (Alexandros Koliatos); writing—review and editing, L.M. and G.A.; supervision, A.K. (Annie Kortsari) and G.A. All authors have read and agreed to the published version of the manuscript.

**Funding:** This research was funded by the Shift2Rail Joint Undertaking under the European Union's Horizon 2020 research and innovation program under grant agreement no. 101015145.

**Institutional Review Board Statement:** Not applicable.

**Informed Consent Statement:** Not applicable.

**Acknowledgments:** The views expressed in this paper are purely those of the authors and may not, under any circumstances, represent those of the Shift2Rail Joint Undertaking. None of the funders played any role in the conduct of this review, in the interpretation of its outputs, in the writing of this report, or the decision to submit this article for publication.

**Conflicts of Interest:** The authors declare no conflict of interest.

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
