# Peer review of "The Hyperloop System and Stakeholders: A Review and Future Directions"

_sustainability, doi:10.3390/su13158430_

Round 1

Reviewer 1 Report

Overall the paper looks interesting.

1. I would recommend including and reviewing the following articles in your revised submission:

https://doi.org/10.1016/j.trip.2020.100092

https://doi.org/10.1016/j.ssci.2021.105366

2. Please highlight the novelty of your work in comparison to

https://doi.org/10.3390/app11135951

3. Table 1 focuses only on developments in Europe. Please include other continents as well.

Reviewer 2 Report

>>My comments to the paper titled "The Hyperloop System and Stakeholders: A Review and Future Directions" are as follows:

>>The first part of the abstract looks like an introduction. The Abstract should include the type of problems/questions that the survey is targeting, the objectives that the survey attempts to achieve, and the concluding remarks or outcomes that are extracted from the survey.

>>Again, the Hyperloop system and stakeholders as a problem and the related research questions should be addressed in the Introduction Section along with existing solutions, e.g., the review of [62].

>>The first part of the methodology is adequate. However, the analysis methods and expected outcomes are missing.

>>The two first research questions of the paper are directly related to the stakeholders, and the other two questions are related to the development of the Hyperloop. Now, more important questions might need to be addressed in this paper like practicality, technical risk, safety, and actual energy efficiency of this technology because such questions will influence the stakeholders' decisions regarding the investment in this technology.

>>Please change the caption of Figure 1 " Figure 1. Stakeholder percentage related to hyperloop per country. Source: Prepared by authors." to Figure 5.

>>I advise making the background color of Figures 2, 3, 5, and 6 plain white background.

>>"Source: Prepared by authors." is unnecessary to include as long as the content 100% belongs to the authors, then no citation is required.

>>The results gave a description of the Hyperloop publications with less deep analysis of the contents of these publications.

>>The results should include the concluding remarks and limitations of this study.

>>The conclusion section looks fine. However, future work is important and needs to be included.

>>Most of the references are adequate, but there are many general reports along with the research articles.

Round 2

Reviewer 1 Report

The authors have addressed all my comments